# Generalized laws of thermodynamics in the presence of correlations

Manabendra N. Bera [1,2], Arnau Riera[1,2], Maciej Lewenstein[1,3] & Andreas Winter [3,4]

The laws of thermodynamics, despite their wide range of applicability, are known to break down when systems are correlated with their environments. Here we generalize thermodynamics to physical scenarios which allow presence of correlations, including those where strong correlations are present. We exploit the connection between information and physics, and introduce a consistent redefinition of heat dissipation by systematically accounting for the information flow from system to bath in terms of the conditional entropy. As a consequence, the formula for the Helmholtz free energy is accordingly modified. Such a remedy not only fixes the apparent violations of Landauer's erasure principle and the second law due to anomalous heat flows, but also leads to a generally valid reformulation of the laws of thermodynamics. In this information-theoretic approach, correlations between system and environment store work potential. Thus, in this view, the apparent anomalous heat flows are the refrigeration processes driven by such potentials.

[1] ICFO – Institut de Ciències Fotòniques, The Barcelona Institute of Science and Technology, ES-08860 Castelldefels, Spain. [2] Max-Planck-Institut für Quantenoptik, D-85748 Garching, Germany. [3] ICREA – Institució Catalana de Recerca i Estudis Avançats, Pg. Lluis Companys 23, ES-08010 Barcelona, Spain. [4] Departament de Física: Grup d'Informació Quàntica, Universitat Autònoma de Barcelona, ES-08193 Bellaterra, Spain. M.N. Bera and A. Riera contributed equally to this work. M. Lewenstein and A. Winter jointly supervised this work. Correspondence and requests for materials should be addressed to A.R. (email: Arnau.Riera@icfo.eu)

Thermodynamics is one of the most successful physical theories ever formulated. Though it was initially developed to deal with steam engines and, in particular, the problem of conversion of heat into mechanical work, it has survived even after the scientific revolutions of relativity and quantum mechanics. Inspired by resource theories, recently developed in quantum information, a renewed effort has been made to understand the foundations of thermodynamics in the quantum domain[1–11], including its connections to statistical mechanics[12–14] and information theory[15–25]. However, all these approaches assume that the system is initially uncorrelated from the bath. In fact, in the presence of correlations, the laws of thermodynamics can be violated. In particular, when there are inter-system correlations, phenomena such as anomalous heat flows from cold to hot baths[26], and memory erasure accompanied by work extraction instead of heat dissipation[24] become possible. These two examples indicate a violation of the second law in its Clausius formulation, and the Landauer's principle of information erasure[15] respectively. Due to the interrelation between the different laws of thermodynamics, the zeroth law and the first law can also be violated (see Supplementary Note 4 for simple and explicit examples of these violations).

The theory of thermodynamics can be summarized in its three main laws. The zeroth law introduces the notion of thermal equilibrium as an equivalence relation of states, where temperature is the parameter that labels the different equivalence classes. In particular, the transitive property of the equivalence relation implies that if a body A is in equilibrium with a body B, and B is with a third body C, then A and C are also in equilibrium. The first law assures energy conservation. It states that in a thermodynamic process not all of energy changes are of the same nature and distinguishes between work, the type of energy that allows for "useful" operations as raising a weight, and its complement heat, any energy change which is not work. Finally, the second law establishes an arrow of time. It has several formulations and perhaps the most common one is the Clausius statement, which reads: No process is possible whose sole result is the transfer of heat from a cooler to a hotter body. Such a restriction not only introduces the fundamental limit on how and to what extent various forms of energy can be converted to accessible mechanical work, but also implies the existence of an additional state function, the entropy, which has to increase. There is also the third law of thermodynamics; we shall, however, leave it out of the discussion, as it is beyond immediate context of the physical scenarios considered here.

Although the laws of thermodynamics were developed phenomenologically, they have profound implications in information theory. The paradigmatic example is the Landauer erasure principle, which states: "Any logically irreversible manipulation of information, such as the erasure of a bit or the merging of two computation paths, must be accompanied by a corresponding entropy increase in non-information-bearing degrees of freedom of the information processing apparatus or its environment"[17]. Therefore, an erasing operation is bound to be associated with a heat flow to the environment.

An important feature in the microscopic regime is that the quantum particles can exhibit non-trivial correlations, such as entanglement[27] and other quantum correlations[28]. Thermodynamics in the presence of correlations has been considered only in limited physical situations. It is assumed, in nearly all cases of thermodynamical processes, that system and bath are initially uncorrelated, although correlations may appear in the course of the process. In fact, it has been noted that in the presence of such correlations, Landauer's erasure principle could be violated[15]. Even more strikingly, with strong quantum correlation between two thermal baths of different temperatures, heat could flow from the colder bath to the hotter one[26,29,30].

The impact of inter-system correlations resulting from a strong system-bath coupling and its role in thermodynamics has been studied for some specific solvable models[31–33], and for general classical systems[34,35]. It has been noted that the presence of correlations requires certain adjustments of work and heat to fulfil the second law and the Landauer principle. Also, from an information theoretic perspective, both extractable work from correlations and work cost to create correlations have been studied[25,36–38]. However, in all these works, there is no explanation of how to deal with general correlated scenarios irrespective of where the correlations come from and in systems away from thermal equilibrium.

Here we show that the violations of the laws of thermodynamics (Supplementary Note 4) indicate that correlations between two systems, irrespective of the corresponding marginals being thermal states or not, manifest out-of-equilibrium phenomena. In order to re-establish the laws of thermodynamics, one not only has to look at the local marginal systems, but also the correlations between them. In particular, we start by redefining the notions of heat and work, then establish a generalized Landauer's principle and introduce the generalized Helmholtz free energy. The resulting laws are general in the sense that they rely on the least set of assumptions to formulate thermodynamics: a system, a considerably large thermal bath at well defined temperature, and separable initial and final Hamiltonians. The first two assumptions are obvious. The third assumption is basically required for system's and bath's energies to be well defined (see Supplementary Note 2 for details).

## Results

**Definition of heat**. To reformulate thermodynamics, we start with redefining heat by properly accounting for the information flow and thereby restoring Landauer's erasure principle. In general, heat is defined as the flow of energy from the environment, normally considered as a thermal bath at certain temperature, to a system, in some way different from work. Work, on the other hand, is quantified as the flow of energy, say to a bath or to an external agent, that could be extractable (or accessible). Consider a thermal bath with Hamiltonian $H_B$ and at temperature $T$ represented by the Gibbs state $\rho_B = \tau_B = \frac{1}{Z_B} \exp\left(\frac{-H_B}{kT}\right)$, where $k$ is the Boltzmann constant, and $Z_B = \text{Tr}\left[\exp\left(\frac{-H_B}{kT}\right)\right]$ is the partition function. The degrees of freedom in B are considered to be a part of a large thermal super-bath, at temperature $T$. Then, for a process that transforms the thermal bath $\rho_B \rightarrow \rho'_B$ with the fixed Hamiltonian $H_B$, the heat transfer to the bath is quantified (see Supplementary Note 1) as

$$\Delta Q = -kT \Delta \mathcal{S}_B, \tag{1}$$

where $\Delta \mathcal{S}_B = \mathcal{S}(\rho'_B) - \mathcal{S}(\rho_B)$ is the change in bath's von Neumann entropy, $\mathcal{S}(\rho_B) = -\text{Tr}\left[\rho_B \log_2 \rho_B\right]$. Note that $\rho'_B$ is not in general thermal. In fact, the work stored in the bath is $\Delta F_B$, where $F(\rho_B) = E(\rho_B) - kT\mathcal{S}(\rho_B)$ is the Helmholtz free energy, with $E(\rho_B) = \text{Tr}(H_B \rho_B)$. Heat expressed in Eq. (1) is the correct quantification of heat (for further discussion see Supplementary Note 1), which can be justified in two ways. First, it has a clear information theoretic interpretation, which accounts for the information flow to the bath. Second, it is the flow of energy to the bath other than work and, with the condition of entropy preservation, any other form of energy flow to the bath will be stored as extractable work, and thus will not converted into heat. The process-dependent character of heat as defined here can be seen from the fact that it cannot be written as a difference of state

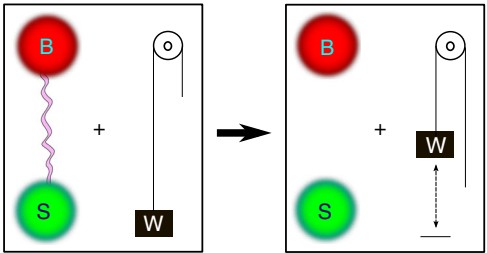

**Fig. 1** Correlations as a work potential. Correlations can be understood as a work potential, as quantitatively expressed in Eq. (4)

functions of the system. In the Supplementary Note 1, this issue is discussed and the sources of irreversibility, i.e., the reasons for not saturating the Clausius inequality, are re-examined.

The transformations considered in our framework are entropy-preserving operations. More explicitly, given a system-bath setting initially in a state $\rho_{SB}$, in which the reduced state of the system $\rho_S$ is arbitrary while $\rho_B$ is thermal, we consider transformations $\rho'_{SB} = \Lambda(\rho_{SB})$ such that the von Neumann entropy is unchanged, i.e., $S(\rho'_{SB}) = S(\rho_{SB})$. The Hamiltonians of the system and the bath are the same before and after the transformation $\Lambda(\cdot)$. Note that we do not demand energy conservation, rather assuming that a suitable battery takes care of that. In fact, the work cost of such an operation $\Lambda(\cdot)$ is quantified by the global internal energy change $\Delta W = \Delta E_S + \Delta E_B$. Another comment to make is that we implicitly assume a bath of unbounded size; namely, it consists of the part $\rho_B$ of which we explicitly track the correlations with S, but also of arbitrarily many independent degrees of freedom. Also, we are implicitly considering always the asymptotic scenario of $n \to \infty$ copies of the state in question ("thermodynamic limit"). These operations are general and include any process and situation in standard thermodynamics involving a single bath. It is the result of abstracting the essential elements of thermodynamic processes: existence of a thermal bath and global entropy preservation operations.

In extending thermodynamics in correlated scenarios and linking thermodynamics with information, we consider the quantum conditional entropy as the natural quantity to represent information content in the system as well as in the correlations. For a joint system-bath state $\rho_{SB}$, the information content in the system S, given all the information available in the bath B at temperature $T$, is quantified by the conditional entropy $S(S|B) = S(\rho_{SB}) - S(\rho_B)$. It vanishes when the joint system-environment state is perfectly classically correlated and can even become negative in the presence of entanglement.

**Generalized second law of information**. With quantum conditional entropy, the generalized second law of information can be stated as follows. For an entropy-preserving operation $\rho'_{SB} = \Lambda^{SB}(\rho_{SB})$, with the reduced states before (after) the evolution denoted $\rho_S$ ($\rho'_S$) and $\rho_B$ ($\rho'_B$), respectively, we have

$$\Delta S_B = -\Delta S(S|B), \qquad (2)$$

where $\Delta S_B = S(\rho'_B) - S(\rho_B)$ is the change in (von Neumann) entropy of the bath, and $\Delta S(S|B) = S(S'|B') - S(S|B)$ is the change in conditional entropy of the system. Note that in the presence of initial correlations, the informational second law could be violated if one considers only system entropy (see Supplementary Note 3).

Let us point out that the conditional entropy of the system for a given bath is also used in ref. 24 in the context of erasing. There, it is shown that the conditional entropy quantifies the amount of

work necessary to erase quantum information. The formalism in ref. 24 considers energy preserving but non-entropy-preserving operations and that perfectly enables to quantify work. In contrast, in our formalism, as we attempt to quantify heat in connection with information flow, it is absolutely necessary to guarantee information conservation, thereby restrict ourselves to entropy-preserving operations. This leads us to quantify heat in terms of conditional entropy. Both approaches are different and complement each other. In one, the conditional entropy quantifies work, and on the other, it quantifies heat.

**Generalized Landauer's principle**. The Landauer principle is required to be expressed in terms of conditional entropy of the system, rather than its local entropy. Therefore, the dissipated heat associated to information erasure of a system S connected to a bath B at temperature $T$ by an entropy-preserving operation $\rho'_{SB} = \Lambda^{SB}(\rho_{SB})$, is equal to

$$\Delta Q = kT \Delta S(S|B). \qquad (3)$$

Note that, in complete information erasure, the final conditional entropy vanishes, then $\Delta Q = -kT S(S|B)$.

**Generalized Helmholtz free energy**. We address extraction of work from a system S possibly correlated to a bath B at temperature $T$. Without loss of generality, we assume that the system Hamiltonian $H_s$ is unchanged in the process. Note that the extractable work has two contributions: one comes from system-bath correlations (cf. ref. 25) and the other from the local system alone, irrespective of its correlations with the bath. Here we consider these two contribution separately.

By extracting work from the correlation, we mean any process that returns the system and the bath in the original reduced states, $\rho_S$ and $\rho_B = \tau_B$. The maximum extractable work solely from the correlation, using entropy-preserving operations, is given by

$$W_C = kT \mathcal{I}(S : B), \qquad (4)$$

where $\mathcal{I}(S : B) = S_S + S_B - S_{SB}$ is the mutual information. This is illustrated in Fig. 1. The proof is given by the protocol described below.

*Protocol 1: work extraction from correlations.* Addition of an ancillary system: We attach to $\rho_{SB}$ an ancillary system A with trivial Hamiltonian $H_A = 0$, consisting of $\mathcal{I}(S : B)$ qubits in the maximally mixed state $\tau_A = \left(\frac{\mathbb{1}_2}{2}\right)^{\otimes \mathcal{I}(S:B)}$ (which is thermal!).

Removing the correlations between S and B: By using a global entropy-preserving operation, we make a transformation $\tau_A \otimes \rho_{SB} \to \tau'_A \otimes \rho_S \otimes \rho_B$, where

$$S(AS|B)_{\tau_A \otimes \rho_{SB}} = S(A'S|B)_{\tau'_A \otimes \rho_S \otimes \rho_B}, \qquad (5)$$

and thereby turning the additional state into a pure state $\tau'_A = |\phi\rangle\langle\phi|$ of A, while leaving the marginal system and bath states unchanged. Clearly, the extractable work stored in the correlation is now transferred to the new additional system state $\tau'_L$.

Work extraction: Work is extracted from $\tau'_L$ at temperature $T$, equal to $W_C = \mathcal{I}(S : B)_{\rho_{SB}} kT$.

Disregarding the correlations with a bath at temperature $T$, the maximum extractable work from a state $\rho_s$ is given by $\Delta W_L = F(\rho_S) - F(\tau_S)$, where $\tau_S = \frac{1}{Z_S} \exp\left[-\frac{H_S}{kT}\right]$ is the corresponding thermal state of the system in equilibrium with the bath. Now, in addition to this "local work", we have the work due to correlations, and so the total extractable work $\Delta W_S = \Delta W_L + kT \mathcal{I}(S : B)_{\rho_{SB}}$. Note that, for the system alone, the Helmholtz free energy $F(\rho_S) = E_S - kT S_S$. However, in the presence of correlations, it is modified to generalized Helmholtz

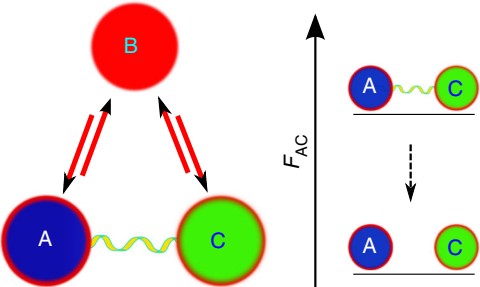

**Fig. 2** Anomalous heat flows. In the presence of correlations, spontaneous heat flows from cold to hot baths are possible[26]. This is an apparent violation of second law, if one ignores the work potential stored in correlation. Otherwise, it is a refrigeration process

**Fig. 3** Violation of the zeroth law. In the presence of correlations, the notion of equilibrium is not an equivalence relation. Consider 3-party state $\rho_B \otimes \rho_{AC}$ with all marginals thermal states. The thermal equilibria $A \leftrightarrows B$ and $B \leftrightarrows C$ imply that A, B and C share the same temperature. But, in the presence of correlations between A and C, that does not assure the equilibrium $A \leftrightarrows C$. Therefore, the transitive property of the equivalence relation is violated. This is justified, on the right, as $F(\rho_{AC}) > F(\rho_A \otimes \rho_C)$. Thus, the generalized zeroth law has to overcome these limitations

free energy, by adding $kT\mathcal{I}(S:B)_{\rho_{SB}}$ to $F(\rho_S)$, as

$$\mathcal{F}(\rho_{SB}) = E_S - kT\mathcal{S}(S|B). \tag{6}$$

Unlike the traditional free energy, the generalized free energy is not only a state function of the system S, but also of those degrees of freedom of the bath correlated with it. This is an unavoidable feature of the generalized formalism. Therefore, for a system-bath state $\rho_{SB}$, maximum extractable work from the system can be given as $\Delta W_S = \mathcal{F}(\rho_{SB}) - \mathcal{F}(\tau_S \otimes \tau_B)$, where $\mathcal{F}(\tau_S \otimes \tau_B) = F(\tau_S)$. Then, for a transformation, for which initial and final states are $\rho_{SB}$ and $\sigma_{SB}$, respectively, the maximum extractable work from the system, is $\Delta W_S = -\Delta\mathcal{F} = \mathcal{F}(\rho_{SB}) - \mathcal{F}(\sigma_{SB})$. We observe that all this is of course consistent with what we know from situations with an uncorrelated bath. Indeed, we can simply make the conceptual step of calling SB "the system", allowing for arbitrary correlations between S and B, with a suitable infinite bath B' that is uncorrelated from SB. Then, the free energy as we know it is $F(\rho_{SB}) = E_S - kT\mathcal{S}(S|B) + E_B - kT\mathcal{S}(\tau_B)$, where the first term is the modified free energy in Eq. (6), and the second term is the free energy of the bath in its thermal state. As the latter cannot become smaller in any entropy-preserving operation, the maximum extractable work is $-\Delta\mathcal{F}$.

**Generalized laws of thermodynamics**. Now, equipped with the proper definition of heat (as in Eq. (3)) and work (based on generalized free energy in Eq. (6)) in the presence of correlations, we put forward the generalized laws of thermodynamics.

We start with generalized first law, which states: given an entropy-preserving operation $\rho_{SB} \rightarrow \rho'_{SB}$, the distribution of the change in the system's internal energy into work and heat satisfies

$$\Delta E_S = -(\Delta W_S + \Delta F_B) + (\Delta Q + \Delta F_B), \tag{7}$$

where the heat dissipated to the bath is given by $\Delta Q = -kT\Delta\mathcal{S}(S|B)$, the maximum extractable work from the system is $\Delta W_S = -(\Delta E_S - kT\Delta\mathcal{S}(S|B))$, and the work performed on the bath is $\Delta F_B = \Delta E_B - kT\Delta\mathcal{S}_B \geqslant 0$.

The quantity $\Delta W_S = -(\Delta E_S - kT\Delta\mathcal{S}(S|B))$ was shown to be the maximum extractable work, as it is equal to $-\Delta\mathcal{F}_S$. The maximum work $\Delta W_S$ is extracted by thermodynamically reversible processes. Irreversible processes require that some work is performed on the bath $\Delta F_B > 0$ followed by an equilibration process, which happens due to spontaneous relaxation of the bath. Such amount of work is transformed into heat and hence cannot be accessed any more. Note that such an equilibration process is not entropy preserving[12] which is not allowed in our setup. The entropy production of such relaxation is precisely $\Delta F_B/T$, and in that case heat flow from the bath is exactly equal to the decrease of its internal energy.

In this new approach, the second law is also modified. The Clausius statement of the generalized second law states that no process is possible whose sole result is the transfer of heat from a

cooler to a hotter body, where the work potential stored in the correlations, as defined in Eq. (4), does not decrease. To prove it, consider a state transformation $\rho'_{AB} = \Lambda^{AB}(\rho_{AB})$ where $\Lambda^{AB}$ is an entropy-preserving and an energy-non-increasing operation. As the thermal state minimizes the free energy, the final reduced states $\rho'_S$ and $\rho'_B$ have increased their free energy, i.e., $\Delta E_A - kT_A\Delta\mathcal{S}_A \geqslant 0$ and $\Delta E_B - kT_B\Delta\mathcal{S}_B \geqslant 0$, where $T_{A/B}$, $\Delta E_{A/B}$ and $\Delta\mathcal{S}_{A/B}$ are the initial temperatures, changes in internal energy and entropy of the baths, respectively. By adding the former inequalities and considering energy non-increasing, we get $T_A\Delta\mathcal{S}_A + T_B\Delta\mathcal{S}_B \leqslant 0$. Due to the conservation of total entropy, the change in mutual information is simply $\Delta\mathcal{I}(A:B) = \Delta\mathcal{S}_A + \Delta\mathcal{S}_B$, with $\mathcal{I}(A:B) = \mathcal{S}_A + \mathcal{S}_B - \mathcal{S}_{AB}$. This allows us to conclude

$$-\Delta Q_A(T_B - T_A) \geqslant kT_A T_B \Delta\mathcal{I}(A:B), \tag{8}$$

which implies Clausius statement of the generalized second law.

Note that if the initial state $\rho_{AB}$ is correlated, then the change in mutual information could be negative, $\Delta\mathcal{I}(A:B) \leqslant 0$, and $-\Delta Q_A(T_B - T_A) \leqslant 0$. Note that for $T_A \leqslant T_B$ and $\Delta\mathcal{I}(A:B) \leqslant 0$, there could be a heat flow from the cold to the hot bath $\Delta Q_A \geqslant 0$, i.e., an apparent anomalous heat flow. From our new perspective, we interpret the anomalous heat flow as a refrigeration driven by the work potential stored in correlations. In this case, it is interesting to determine its coefficient of performance $\eta_{cop}$, that from Eq. (8) leads, with the work performed on the hot bath $\Delta W_C(T_B) = -kT_B\Delta\mathcal{I}(A:B)$, to

$$\eta_{cop} := \frac{\Delta Q_A}{\Delta W_C(T_B)} \leqslant \frac{T_A}{T_B - T_A}, \tag{9}$$

which is nothing else than the Carnot coefficient of performance (Fig. 2). Note that we have taken the work value of the correlations $W_C$ with respect to the hot bath $T_B$. This is due to the fact that for this refrigeration process the hot bath is the one acting as a reservoir.

Equation (9) is a nice reconciliation with traditional thermodynamics. The Carnot coefficient of performance is a consequence of the fact that reversible processes are optimal, otherwise the perpetual mobile could be build by concatenating a "better" process and a reversed reversible one. Hence, it is natural that the refrigeration process driven by the work stored in the correlations preserves Carnot statement of second law.

Now, we reconstruct the zeroth law which can be violated in the presence of correlations as shown in Fig. 3. To do this, we redefine the notion of equilibrium beyond an equivalence relation when correlations between systems are present. Thus, the generalized zeroth law states that, a collection $\{\rho_X\}_X$ of states is said to be in mutual thermal equilibrium with each other if and only if no work can be extracted from any of their combinations under entropy-preserving operations. This is the case if and only if all the parties $X$ are uncorrelated and each of them is in a thermal state with the same temperature.

## Discussion

Landauer exorcised Maxwell's demon and saved the second law of thermodynamics by taking into account the work potential of information. In this work, we extend this idea to include also the information about the system that is stored in its correlations with the environment. With this approach, we easily resolve the apparent violations of thermodynamics in correlated scenarios, and generalize it by reformulating its zeroth, first, and second laws.

An important remark is that, our generalized thermodynamics is formulated in the asymptotic limit of many copies. A relevant question is how the laws of thermodynamics are expressed for a single system. In our forthcoming paper, we will address these questions by discussing consistent notions of one-shot heat, one-shot Landauer erasure, and of one-shot work extraction from correlations.

**Data availability**. Data sharing not applicable to this article as no data sets were generated or analyzed during the current study.

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

## Acknowledgements

We thank R. Alicki, R. B. Harvey, K. Gawedzki, P. Grangier, J. Kimble, V. Pellegrini, R. Quidant, D. Reeb, F. Schmidt-Kaler, P. Walther and H. Weinfurter for useful discussions and comments in both theoretical and experimental aspects of our work. We acknowledge financial support from the European Commission (FETPRO QUIC (H2020-FET-PROACT-2014 No. 641122), STREP EQuaM and STREP RAQUEL), the European Research Council (AdG OSYRIS and AdG IRQUAT), the Spanish MINECO (grants nos FIS2008-01236, FIS2013-46768-P FOQUS, FISICATEAMO FIS2016-79508-P and Severo Ochoa Excellence Grant SEV-2015-0522) with the support of FEDER funds, the Generalitat de Catalunya (grants nos SGR 874, 875 and 966), CERCA Program/Generalitat de Catalunya and Fundació Privada Cellex. AR also thanks support from the CELLEX-ICFO-MPQ fellowship.

## Author contributions

All authors discussed the results and contributed to the final manuscript.

## Additional information

**Competing interests:** The authors declare no competing financial interests.

