## [Peer Review File · Nature Communications]

Reviewers' comments:

Reviewer #1 (Remarks to the Author):

In the manuscript at hand, the authors sweepingly claim to “universalize” thermodynamics “by reformulating its (...) laws.” A number of well-known algebraic relations between entropic quantities are used to define heat and work, along with formulations of the 0th, 1st, and 2nd laws of thermodynamics in a way that apparently* eliminates possible misinterpretations (*ruling out people’s will to misinterpret should not be underestimated).

There are not many technical remarks that I can make that have not been made already by the previous reviewers. I also believe that any purely technical issues of prior versions have been addressed in the revised version, and from a purely technical, physical point of view the manuscript seems to be correct. I will therefore focus primarily on the conceptual importance of the present manuscript, as I perceive it.

First, let me say that I do believe that the paper is useful in the sense that it collects a number of thermodynamic statements in a unified language that is suitable for a modern approach to quantum thermodynamics based on quantum information. I also agree with the authors that some misconceptions regarding the potential violation of thermodynamic laws may persist in the literature, possibly intentionally, by ignoring classical assumptions such as baths that are not correlated with the system. That being said, it seems to me that while what the authors claim here is perfectly reasonable, it is also not entirely unknown or surprising to the community, as all of the previous reviewers have pointed out.

The authors have, in my opinion, replied to all of the technical remarks of previous reviewers, but, again, the claims regarding the conceptual novelty of their results are somewhat overstated. For instance, the claim that “...our analysis shows that one could extract work from correlations,” feels strange given that the title of their reference [25] is “Extractable Work from Correlations”. In particular, in conjunction with the claim “In fact, even to quantify the work cost of creating arbitrary correlation, using most efficient transformation, one has to rely on the protocol presented in our work.” given that the works M. Huber et al 2015 New J. Phys. 17 065008 and D. Bruschi, et al Phys. Rev. E 91, 032118 have been pointed out already by a previous reviewer. In the latter of these papers, the equivalent of Eq. (4) is shown to be the lower bound for the work cost of creating these correlations in the first place, with equality when no correlations are created with third systems

(such as the superbath mentioned by the authors). It is then trivial to see that the extractable work from the joint system is bounded from above by the expression given in Eq. (4). In their reply, the authors correctly remark that both of the mentioned papers consider the creation of correlations in initially uncorrelated systems. But, first, that does not seem to make these results less relevant to what is claimed in the present paper in my opinion, so I don't see how it matters that "we have cited a very related work from the same authors [25]". Second, there is also another work (Phys. Rev. E 93, 042135 (2016)) discussing the work cost of creating correlations in initially correlated systems that may be relevant here.

So, while the results of the paper are interesting in principle, they do not (and, frankly, cannot) live up to the extremely high expectations generated by the presentation, suggesting that the manuscript single-handedly reinvents thermodynamics. Instead, it seems that the authors consistently combine previous approaches to then carefully phrase thermodynamic laws in a way that is maybe less likely to be misinterpreted.

I hence believe that this could be a nice paper in a different journal, for which I would recommend (i) not to hide all the technical results in the supplemental material, (ii) to be slightly more humble in the assessment of the contribution of this paper relative to existing work, (iii) check the manuscript for language (although the main narrative is well written, a number of prepositions are used incorrectly, some articles and verbs are missing). Overall, I conclude that although this paper is a reasonably nice contribution to the field, it is in my view not sufficiently novel or ground-breaking to warrant publication in Nature Communications.

Reviewer #2 (Remarks to the Author):

It is my opinion that this is an interesting and well written paper, which meets the criteria for publication in Nature Communications in that it contains novel results that are going to be important for scientists in the field of quantum thermodynamics.

In particular, I think that the authors have answered the criticisms of the three referees and that the paper is now quite easily readable and free from the ambiguities that have been signalled already. There is only one point in which I would keep the position of previous Referees, namely that of calling this formulation "universal".

Since there are many conceivable generalizations (including, e.g. the single shot case mentioned by the authors themselves, or that with a finite environment in a thermal state, or with more general transformation allowed), I would suggest to use "generalized" instead of "universal".

I understand, however, that this can be more a matter of taste than anything else, so this remark should be intended as a strong suggestion rather than as an imperative for publication.

Besides this optional remark, I would ask the authors to comment on the definition of heat in eq. (1) in the following sense:

from the paper, it seems that the transformation considered in can be a generic finite one (but for the entropy preservation), so that both heat and entropy changes are finite quantities and the final state ρ_B' is an out of equilibrium state.

If I'm guessing right, the authors are imagining that the bath is so large that the change induced by the transformation is, in fact, infinitesimal, so that ρ_B' is close to ρ_B in some sense. If I'm right, this should be stated explicitly; while, if I'm wrong, how can you justify the use of the initial temperature T in the definition of the free energy of the evolved state, $F(\rho_B')$ and in the relation $\Delta Q = -kT \Delta S$?

Once this point is clarified, I will be happy to suggest the acceptance of the manuscript.

Reviewer #3 (Remarks to the Author):

Initial correlation is not the only hang up in extending thermodynamics to out-of-equilibrium systems. Moreover, while the information-energy conversion (Landauer principle) is one of the most important discoveries of 20th century, it by no means encapsulates all of the thermodynamics. Thus, I find the usage of term "universal" to be a slight overreach.

What the authors have done is to generalise the laws of thermodynamics, as we know them, to allow for initial correlations between the system and the environment. This is quite an achievement indeed, and therefore I am in support of this paper. However, I do think that the authors must replace the term universal with something much more reasonable.

What is unclear to me is the assumption on the total initial state. Is it thermal? Or is the reduced state of the bath thermal? This point should be clearly explained in the main text itself.

I have also reviewed the authors' reply to previous referees.

Referee 1 recommends several papers for the authors to cite. In each case, in my opinion, authors defended their choice.

Referee 2 raises several concerns about interpretation of the "new" laws of thermodynamics. I think of the concerns are reasonable, but the authors address them clearly.

Referee 3 claims that normal thermodynamics apply at the system bath level and thus at system level too. However, I don't think that implies much about the subsystems in strong coupling limit. Initial correlations imply strong coupling.

Perhaps it is worth noting that the present approach is more in line with "information thermodynamics" rather than "traditional thermodynamics." Arguably the two are the same, but as the two communities tend not to agree even at the level of definitions, they may not be the same theories.

Overall I find this to be a nice contribution and I recommend it for publication.

Reply to the referees:

Color code:

Text in black are the referee comments.

Text in **blue** our reply to the referees.

Reviewer #1 (Remarks to the Author):

In the manuscript at hand, the authors sweepingly claim to “universalize” thermodynamics “by reformulating its (...) laws.” A number of well-known algebraic relations between entropic quantities are used to define heat and work, along with formulations of the 0th, 1st, and 2nd laws of thermodynamics in a way that apparently* eliminates possible misinterpretations (*ruling out people’s will to misinterpret should not be underestimated).

There are not many technical remarks that I can make that have not been made already by the

previous reviewers. I also believe that any purely technical issues of prior versions have been addressed in the revised version, and from a purely technical, physical point of view the manuscript seems to be correct. I will therefore focus primarily on the conceptual importance of the present manuscript, as I perceive it.

First, let me say that I do believe that the paper is useful in the sense that it collects a number of thermodynamic statements in a unified language that is suitable for a modern approach to quantum thermodynamics based on quantum information. I also agree with the authors that some misconceptions regarding the potential violation of thermodynamic laws may persist in the literature, possibly intentionally, by ignoring classical assumptions such as baths that are not correlated with the system. That being said, it seems to me that while what the authors claim here is perfectly reasonable, it is also not entirely unknown or surprising to the community, as all of the previous reviewers have pointed out.

The authors have, in my opinion, replied to all of the technical remarks of previous reviewers, but, again, the claims regarding the conceptual novelty of their results are somewhat overstated. For instance, the claim that "...our analysis shows that one could extract work from correlations," feels strange given that the title of their reference [25] is "Extractable Work from Correlations". In particular, in conjunction with the claim "In fact, even to quantify the work cost of creating arbitrary correlation, using most efficient transformation, one has to rely on the protocol presented in our work." given that the works M. Huber et al 2015 New J. Phys. 17 065008 and D. Bruschi, et al Phys. Rev. E 91, 032118 have been pointed out already by a previous reviewer. In the latter of these papers, the equivalent of Eq. (4) is shown to be the lower bound for the work cost of creating these correlations in the first place, with equality when no correlations are created with third systems (such as the superbath mentioned by the authors). It is then trivial to see that the extractable work from the joint system is bounded from above by the expression given in Eq. (4). In their reply, the authors correctly remark that both of the mentioned papers consider the creation of correlations in initially uncorrelated systems. But, first, that does not seem to make these results less relevant to what is claimed in the present paper in my opinion, so I don't see how it matters that "we have cited a very related work from the same authors [25]". Second, there is also another work (Phys. Rev. E 93, 042135 (2016)) discussing the work cost of creating correlations in initially correlated systems that may be relevant here.

We thank the referee for finding our work interesting and suggesting several references that we have included in our revised manuscript. The novelty of our work lies in the fact that it can be applied to general situations irrespective of where the correlations come from and in systems away from thermal equilibrium. In addition, we show how these correlation can be incorporated to thermodynamics and how its laws can be generalized.

So, while the results of the paper are interesting in principle, they do not (and, frankly, cannot) live up to the extremely high expectations generated by the presentation, suggesting that the manuscript single-handedly reinvents thermodynamics. Instead, it seems that the authors consistently combine previous approaches to then carefully phrase thermodynamic laws in a way that is maybe less likely to be misinterpreted.

I hence believe that this could be a nice paper in a different journal, for which I would recommend (i) not to hide all the technical results in the supplemental material, (ii) to be slightly more humble in the assessment of the contribution of this paper relative to existing work, (iii)

check the manuscript for language (although the main narrative is well written, a number of prepositions are used incorrectly, some articles and verbs are missing). Overall, I conclude that although this paper is a reasonably nice contribution to the field, it is in my view not sufficiently novel or ground-breaking to warrant publication in Nature Communications.

With the argument given above and the changes made in the revised manuscript, we hope that our work is now suitable for publication in Nature Communications.

Reviewer #2 (Remarks to the Author):

It is my opinion that this is an interesting and well written paper, which meets the criteria for publication in Nature Communications in that it contains novel results that are going to be important for scientists in the field of quantum thermodynamics.

In particular, I think that the authors have answered the criticisms of the three referees and that the paper is now quite easily readable and free from the ambiguities that have been signalled already. There is only one point in which I would keep the position of previous Referees, namely that of calling this formulation "universal".

Since there are many conceivable generalizations (including, e.g. the single shot case mentioned by the authors themselves, or that with a finite environment in a thermal state, or with more general transformation allowed), I would suggest to use "generalized" instead of "universal". I understand, however, that this can be more a matter of taste than anything else, so this remark should be intended as a strong suggestion rather than as an imperative for publication.

We thank the referee for finding our work important, novel and well written such that it deserves to be published in Nature Communications. Following the referee suggestion we have replaced the term “universal” with “generalized”, and entitled the manuscript as “Generalized Laws of Thermodynamics in Presence of Correlations”.

Besides this optional remark, I would ask the authors to comment on the definition of heat in eq. (1) in the following sense:

from the paper, it seems that the transformation considered in can be a generic finite one (but for the entropy preservation), so that both heat and entropy changes are finite quantities and the final state ρ_B' is an out of equilibrium state.

If I'm guessing right, the authors are imagining that the bath is so large that the change induced by the transformation is, in fact, infinitesimal, so that ρ_B' is close to ρ_B in some sense. If I'm right, this should be stated explicitly; while, if I'm wrong, how can you justify the use of the initial temperature T in the definition of the free energy of the evolved state, $F(\rho_B')$ and in the relation $\Delta Q = -k T \Delta S$?

We thank the referee for pointing this out. He/she is right and the bath is considered to be large. We have clarified this point in the revised manuscript.

Once this point is clarified, I will be happy to suggest the acceptance of the manuscript.

We thank the referee for his/her suggestion. We hope that now the revised manuscript is ready for publication in Nature Communications.

Reviewer #3 (Remarks to the Author):

Initial correlation is not the only hang up in extending thermodynamics to out-of-equilibrium

systems. Moreover, while the information-energy conversion (Landauer principle) is one of the most important discoveries of 20th century, it by no means encapsulates all of the thermodynamics. Thus, I find the usage of term "universal" to be a slight overreach.

What the authors have done is to generalise the laws of thermodynamics, as we know them, to allow for initial correlations between the system and the environment. This is quite an achievement indeed, and therefore I am in support of this paper. However, I do think that the authors must replace the term universal with something much more reasonable.

We thank the referee for finding our work valuable. Following his/her suggestion, we have replaced the term "universal" with "generalized", and entitled the manuscript as "Generalized Laws of Thermodynamics in Presence of Correlations".

What is unclear to me is the assumption on the total initial state. Is it thermal? Or is the reduced state of the bath thermal? This point should be clearly explained in the main text itself.

We thank the referee for raising this point. Indeed the initial bath state is in thermal equilibrium. This issue has been made clear in the revised manuscript.

I have also reviewed the authors' reply to previous referees.

Referee 1 recommends several papers for the authors to cite. In each case, in my opinion, authors defended their choice.

Referee 2 raises several concerns about interpretation of the "new" laws of thermodynamics. I think of the concerns are reasonable, but the authors address them clearly.

Referee 3 claims that normal thermodynamics apply at the system bath level and thus at system level too. However, I don't think that implies much about the subsystems in strong coupling limit. Initial correlations imply strong coupling.

Perhaps it is worth noting that the present approach is more in line with "information thermodynamics" rather than "traditional thermodynamics." Arguably the two are the same, but as the two communities tend not to agree even at the level of definitions, they may not be the same theories.

This is indeed an important observation and we agree with the referee.

Overall I find this to be a nice contribution and I recommend it for publication.

We thank the referee for his/her suggestion. We hope that now the revised manuscript is ready for publication in Nature Communications.